# Fungal mycetoma and pregnancy: An association with costly and difficult management, about a case

**Niare Ndour** [1]*, **Mame Tené Ndiaye**[2], **Assane Diop**[1], **Ramatoulaye Ly**[1], **Coumba Ndiaye**[1], **Mamadou Sarr**[1], **Astou Diouf**[1], **Fatou Diagne**[1], **Fatimata Ly**[1]

1 Department of Dermatology, Hospital Institute of Social Hygiene, Dakar, Senegal, 2 Department of Dermatology, Albert Royer Hospital, Dakar, Senegal

* ndourniar0@gmail.com

**Data Availability Statement:** All data are in the manuscript.

**Funding:** The author(s) received no specific funding for this work.

## Abstract

Mycetomas are endemic diseases in tropical and sub-tropical countries of Africa, Asia and America, mainly affecting rural populations living below the poverty line. We report a particular case of a mycetoma associated with pregnancy whose evolution was good, but at the cost of significant financial expenses. This was a 39-year-old woman who developed a fungal mycetoma due to *Madurella mycetomatis* from the ingunocrural region. The patient had to develop several previous pregnancies on this site of mycetoma, the outcome of which was favorable. The last pregnancy was accompanied by an aggravation of the mycetoma in the form of polyfistulized inflammatory swelling of the right inguino-crural region emitting black grains. Magnetic Resonance Imaging (MRI) of the region showed invasion of the adductor muscles at the level of the root of the thigh on its antero-internal side with no sign of pelvic extension or underlying bone lesion. The patient was treated by surgery associated with antifungal treatment. The evolution was favorable for pregnancy and mycetoma.

## Author summary

Mycetoma is a neglected tropical disease that occurs mainly in the tropics, where it essentially affects people living in the poorest communities. The health, social and economic consequences are numerous. Our observation describes some aspects related to the difficulty of managing mycetoma, particularly in the case of association with pregnancy, with several changes of course, sometimes worsening, sometimes improving and whose cost of care is high.

## Ethics statement

This article is a case report and therefore has not been submitted to the ethics committee, but written patient consent has been obtained. This consent is attached to the submission file for this article.

**Competing interests:** The authors have declared that no competing interests exist.

## Presentation of the case

A woman aged 39 years, married, 05 pregnancies including 3 live births and 2 successive abortions all occurred in the second month, from the north of Senegal (Louga), was followed since 2009 for a fungal mycetoma of the right inguino-crural region with black grains due to *Madurella mycetomatis* whose diagnosis was based on the histopathological aspect and the mycological culture. All five pregnancies had occurred with this mycetoma, three of which had been successful, for each a full-term delivery of an apparently healthy newborn. Table 1 summarizes the evolution of the mycetoma and the pregnancy during their different associations.

This patient consulted for a new mycetoma relapse in the form of a polyfistulized inflammatory swelling of the right inguino-crural region emitting black grains (Fig 1). As part of the extension assessement, a new Magnetic Resonance Imaging (MRI) of the pelvis and the right thigh was performed and showed an invasion of the adductor muscles at the level of the root of the thigh on its anteromedial side without any sign of pelvic extension or underlying bone lesion. A treatment based on itraconazole 200 mg twice a day and terbinafine 500 mg twice a day was initiated while waiting for a surgical treatment. The patient returned to the clinic 2 months later with a progressive pregnancy which led to the discontinuation of itraconazole and terbinafine. Treatment with amoxicillin-clavulanic acid 2g/day was initiated in view of the inflammatory appearance of the lesions, which were suspected of superinfection. Three months after stopping the initial treatment, and at 6 months of pregnancy, the patient was hospitalized for a worsening of the clinical picture with numerous painless nodules located on the inner side of the root of the right thigh and on the pubis associated with homolateral adenopathies and a non-inflammatory edema of the right lower limb taking the cup. Some of these nodules were fistulized emitting large black grains. Complementary examinations showed normochromic normocytic anemia at 8 g/dl; the reticulocyte count was not done. There was no hyperleukocytosis and the platelet count was normal. The C-Reactive Protein (CRP) was slightly positive at 6 mg/l. Retroviral serology and Hbs antigen were negative. The patient was followed in parallel in gynecology and obstetrics with a good progress of the pregnancy at the time of the hospitalization. A surgery was decided after a multidisciplinary consultation including gynecologists-obstetricians, orthopedic surgeons, visceral surgeons and intensive care anesthetists. After a normal preoperative workup, The patient underwent surgery without incident and received treatment with intravenous voriconazole for 6 days, followed by oral treatment for 1 month. The pregnancy progressed normally until term with a normal delivery of the newborn. The evolution of the mycetoma was marked by a complete healing, 3 months after the operation, under itraconazole 400 mg/d with hypertrophy of the anterior and posterior scars (Figs 2 and 3). The patient had to spend 10,687 euros to achieve this result, divided

**Table 1. The evolution of mycetoma and pregnancy during their different associations.**

| Year | Influence of pregnanc on the mycetoma | Local extension on MRI (muscles and/or bone) | Pregnancy outcome | State of health of the newborn | Treatment |
|---|---|---|---|---|---|
| 2009 | worsening | No | Favorable | Good health | Medical |
| 2010 | worsening | No | Favorable | Good health | Medical |
| 2014 | worsening | No | Favorable | Good health | Médical + surgery in 2 steps |
| 2017 | worsening | No | 2 Abortions at 8 SA | | Médical + surgery |
| 2020 (last episode) | worsening | Yes | Favorable | Good health | Médical + surgery |

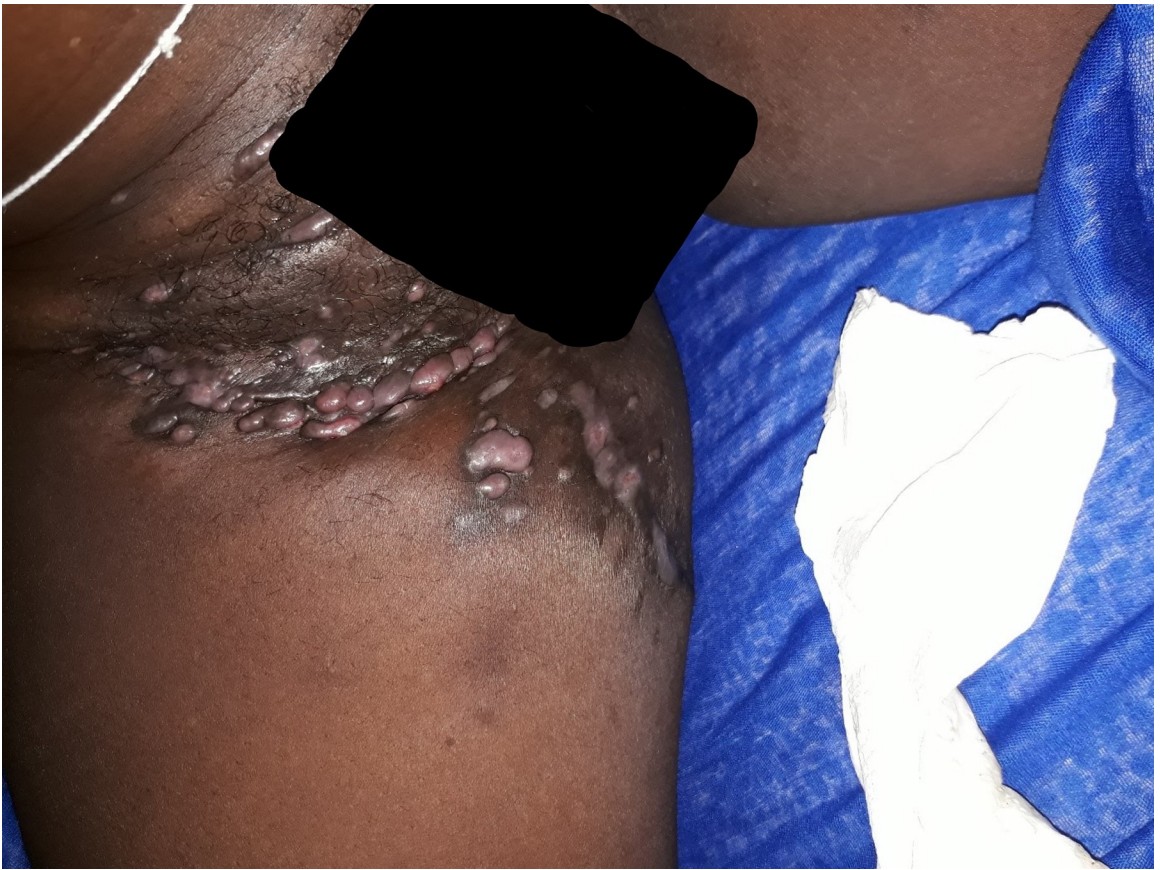

**Fig 1. Outbreak of fungal mycetoma of the right inguinal region with nodules.**

between the purchase of drugs, the costs of surgery, hospitalization costs and payment of complementary examinations without counting travel costs between Louga and Dakar.

## Case discussion

Our observation is particular by the occurrence of a pregnancy in a patient followed for a fungal mycetoma evolving since 2009. Although reported by several authors this is not a very common occurrence. It is characterized by an aggravation of the mycetoma in the form of a new attack. The mechanisms that could explain this aggravation are not clearly identified. This phenomenon is probably related to the immunodeficiency state of pregnant women, probably related to hormonal influence or the decreased cell-mediated immunity with a decreased CD4/CD8 ratio [1,2,3,4]. Other authors suggest that the TH2 cytokine profile may be associated with the development of fungal mycetomas during pregnancy after finding an increase in interleukin Il-10 in these patients [5]. Similarly, Lavalle and al. reported an increase in mycetoma activity in *nocardia brasilensis* in women during pregnancy [6]. These authors believe that there would exist an anti-mycetomic factor in women which would repressed during pregnancy. However, Mohr and Muchmore cite the case of a woman whose evolution of *Allescheria boydii* mycetoma showed periods of remission during three pregnancies and exasperation in the intermediate periods and, during this one, the activity of the mycetoma was manifested during menstruation [7].

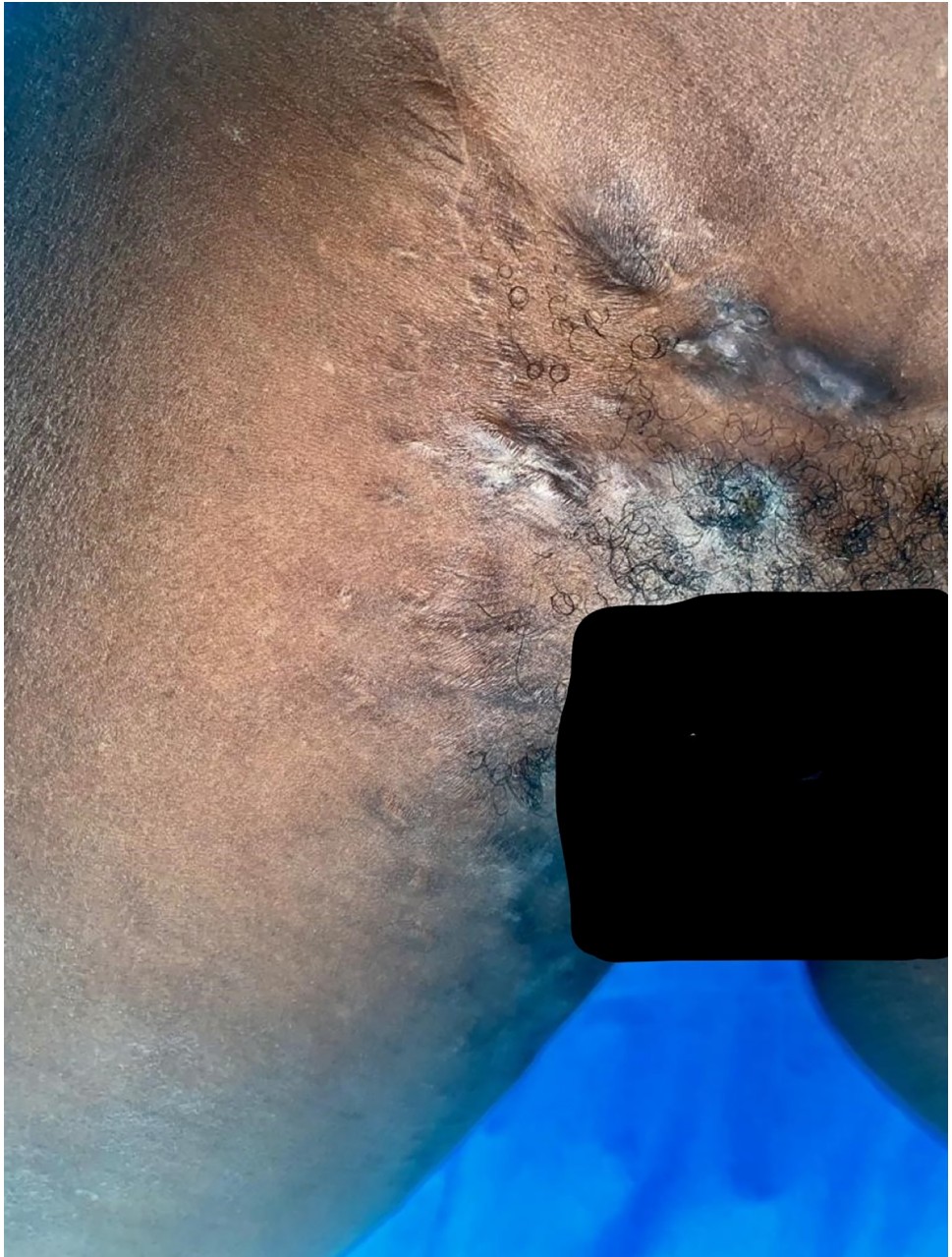

**Fig 2. Good evolution of a pregnancy-associated mycetoma after surgical excision and antifungal treatment with hypertrophic scarring on the anteromedial aspect.**

The other particularity of our observation is that the pregnancy had progressed well in this field of mycetoma despite the inflammatory outbreak with a vaginal delivery of a healthy full-term newborn. Indeed, the main difficulty in the case of mycetoma associated with pregnancy remains therapeutic since the main molecules available for this pathology are contraindicated in pregnancy. Surgical intervention was the best therapeutic option in our patient because it reduced the extension of the lesions. On the other hand, voriconazole, first used by intravenous injection and then given orally, would also have helped to control the outbreak. Voriconazole is a broad-spectrum antifungal agent indicated for the treatment of invasive candidiasis

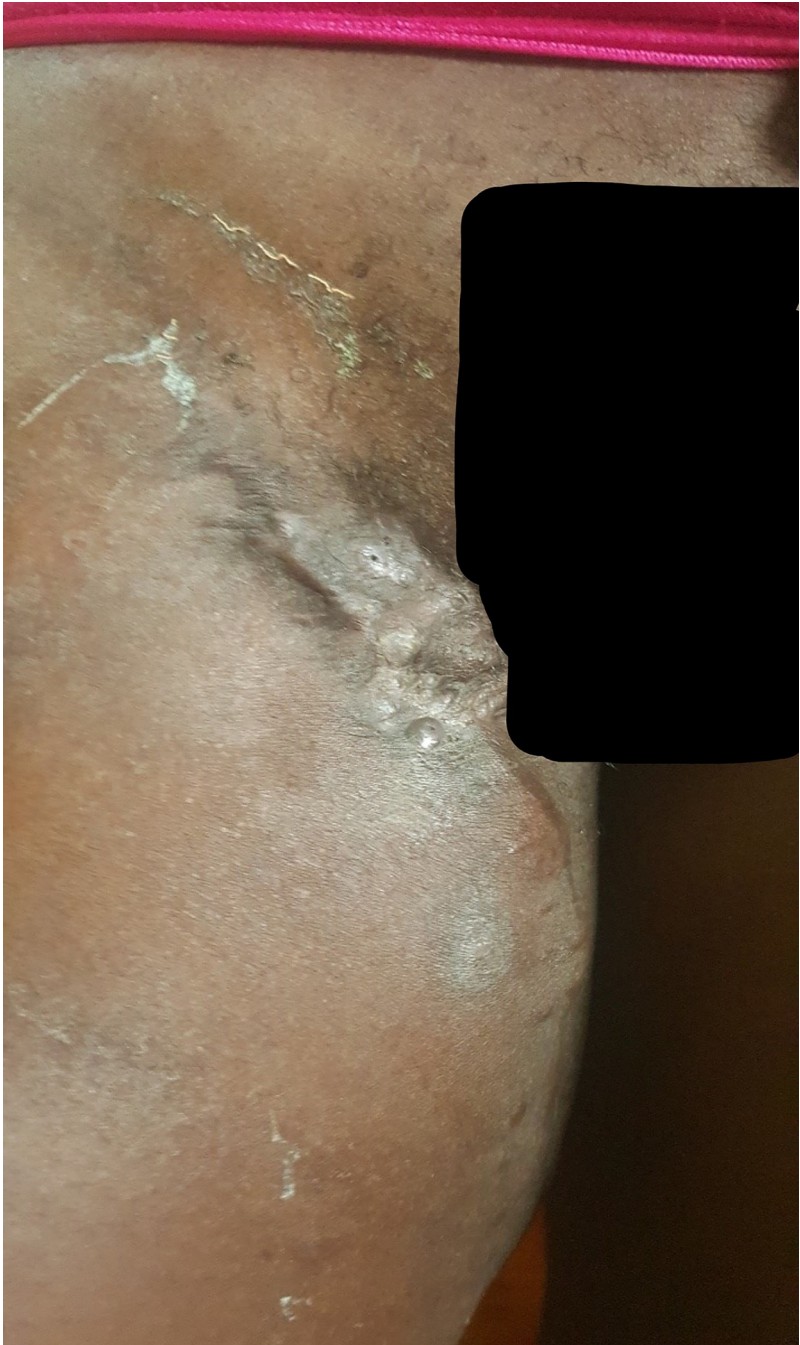

**Fig 3. Good evolution of a pregnancy-associated mycetoma after surgical excision and antifungal treatment with hypertrophic scarring on the root of the thigh.**

resistant to fluconazole, severe fungal infections with *Scedosporium sp* or *Fusarium spp*. A favorable evolution of a *Scedosporium apiospermum* mycetoma under voriconazole has been reported [7].

The evolution of the mycetoma was favorable 6 months after delivery under itraconazole 400 mg/d characterized by an absence of grain emission, a complete healing of the lesions at the cost of some hypertrophic scars. However, the optimal duration of antifungal treatment

**Table 2. Prices of the main drugs used in the treatment of fungal mycetoma.**

| Molecule | Number of tablets per box | Price of one box (euros) | Dosage During Mycetoma | Cost of one month of treatment (euros) |
|---|---|---|---|---|
| Itraconazole 100 mg cp | 30 | 30,83 | 400 mg/d | 217,93 |
| Terbinafine 250 mg capsule | 28 | 21,37 | 1000 mg/d | 151,14 |
| Voriconazole 200 mg cp | 28 | 438,16 | 200 to 400 mg/d | 453,43 to 906,87 |

remains poorly codified and a follow-up of several years is essential before being able to affirm the cure of an eumycetoma [8,9].

The overall cost of our patient's treatment is estimated at 10,687 euros, which is considerable compared to her much lower income, forcing her to take out bank loans and sell most of her material goods. This financial cost can be explained on the one hand firstly by the chronic and recurrent nature of fungal mycetoma requiring long term treatments and on the other hand by the high price of most of the antifungal drugs used during this pathology, but also the cost of the often delicate surgical interventions, the management of the terrain constituted by the pregnancy. Table 2 gives an idea of the cost of one month of each of the main drugs used for the treatment of fungal mycetoma.

## Conclusion

The association of mycetoma and pregnancy is rarely described. The data are discordant on the consequences of this association with sometimes an improvement, sometimes an aggravation. Our case was singled out by a worsening of the mycetoma during all pregnancies contrasting with the good maternal-fetal prognosis in the majority of cases. However, this association poses therapeutic difficulties linked not only to the high cost of the drugs but also to the fact that most of these drugs are contraindicated. In many countries, the high cost of management of neglected tropical skin diseases is linked to the high cost of drugs. Hence the importance of setting up a large-scale health policy in order to cope with the burden of mycetoma.

## Author Contributions

**Conceptualization:** Niare Ndour, Mame Tené Ndiaye, Fatimata Ly.

**Data curation:** Niare Ndour, Ramatoulaye Ly.

**Formal analysis:** Niare Ndour, Mame Tené Ndiaye, Ramatoulaye Ly.

**Investigation:** Niare Ndour, Mame Tené Ndiaye, Assane Diop, Ramatoulaye Ly, Coumba Ndiaye, Mamadou Sarr, Astou Diouf, Fatou Diagne, Fatimata Ly.

**Methodology:** Mame Tené Ndiaye, Fatimata Ly.

**Supervision:** Mame Tené Ndiaye, Assane Diop, Ramatoulaye Ly, Fatimata Ly.

**Validation:** Fatimata Ly.

**Writing – original draft:** Niare Ndour, Mame Tené Ndiaye, Ramatoulaye Ly.

**Writing – review & editing:** Niare Ndour, Mame Tené Ndiaye, Assane Diop, Ramatoulaye Ly, Coumba Ndiaye, Mamadou Sarr, Astou Diouf, Fatou Diagne, Fatimata Ly.

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
