## [Decision Letter · Decision Letter 0]

24 Jun 2023

Dear Dr. NDOUR,

Thank you very much for submitting your manuscript "Fungal mycetoma and pregnancy: an association with costly and difficult management, about a case" for consideration at PLOS Neglected Tropical Diseases. As with all papers reviewed by the journal, your manuscript was reviewed by members of the editorial board and by several independent reviewers. In light of the reviews (below this email), we would like to invite the resubmission of a significantly-revised version that takes into account the reviewers' comments. 

We cannot make any decision about publication until we have seen the revised manuscript and your response to the reviewers' comments. Your revised manuscript is also likely to be sent to reviewers for further evaluation.

Sincerely,

Marcio L Rodrigues

Section Editor

Reviewer's Responses to Questions

**Key Review Criteria Required for Acceptance?**

**Methods**

-Are the objectives of the study clearly articulated with a clear testable hypothesis stated?

-Is the study design appropriate to address the stated objectives?

-Is the population clearly described and appropriate for the hypothesis being tested?

-Is the sample size sufficient to ensure adequate power to address the hypothesis being tested?

-Were correct statistical analysis used to support conclusions?

-Are there concerns about ethical or regulatory requirements being met?

Reviewer #1: This is a single case

Was the M.mycetomatis identified by culture or by molecular methods ? 

Can you be more specific about the surgery ie was this visually complete excision. What doses of voriconazole were used and when was the patient changed to itraconazole

In Table 1 what does thrust mean ? Worsening ??

Reviewer #2: The microbiologic identification of M. mycetomatis is not included

The authors do not explained the reason to stop antifungal therapy and changing to antibiotics

**Results**

-Does the analysis presented match the analysis plan?

-Are the results clearly and completely presented?

-Are the figures (Tables, Images) of sufficient quality for clarity?

Reviewer #1: "The patient had to develop several previous pregnancies on this site of mycetoma, the outcome of which was favourable"

Does this mean that the patient had been pregnant several times before in the presence of the same mycetoma but had not developed any complications ? If so this isn’t what is said in the discussion where it is implied that the patients’ mycetoma was aggravated in all pregnancies

Chain Reactive Protein – C-reactive Protein

"Only a few cases have been reported in the literature concerning this association". I would be more careful here as you haven’t carried out a full literature review and there are more cases. So this is a real and significant association. I would change this to ….Cases have been reported in the literature concerning this association. You again say this is a rare occurrence in the discussion – I would rephrase this to say Although reported by several authors this is not a very common occurrence 

This author believes - there are 9 authors !

 Voriconazole, first injected. Presumably this was by standard iv infusion rather than direct infiltration of the lesion

Table 2 

Is there a missing drug in the first column ? Also some of the data is repeated

Reviewer #2: Table 1 is not necessary and should be substituted by a time lining 

Table 3 is not necessary

**Conclusions**

-Are the conclusions supported by the data presented?

-Are the limitations of analysis clearly described?

-Do the authors discuss how these data can be helpful to advance our understanding of the topic under study?

-Is public health relevance addressed?

Reviewer #1: Conclusion. You might point out that this drug cost problem is a very real one in many countries where the best management of Skin NTDs is compromised by the high cost of medications - in this case antifungals

Reviewer #2: The authors should focus their discussion in the clinical and radiological outcome and especially the use of Antifungal drugs during pregnancy. 

In my opinion, advocacy related to therapy costs is non relevant.

**Editorial and Data Presentation Modifications?**

Reviewer #1: (No Response)

Reviewer #2: (No Response)

**Summary and General Comments**

Reviewer #1: Overall the use of English should be checked and changed were necessary

Reviewer #2: The authors should change the focus of their manuscript.

They should discuss the antingungal drugs during pregnancy

The costs of antifungals and or the therapy in African countries is not scientifically interessante, but they could resume this issues in a single sentence

PLOS authors have the option to publish the peer review history of their article (what does this mean?). If published, this will include your full peer review and any attached files.

Reviewer #1: No

Reviewer #2: Yes: Flavio Queiroz-Telles
---

## [Editor Report · Decision Letter 1]

15 Aug 2023

Dear Dr. NDOUR,

Thank you very much for submitting your manuscript "Fungal mycetoma and pregnancy: an association with costly and difficult management, about a case" for consideration at PLOS Neglected Tropical Diseases. As with all papers reviewed by the journal, your manuscript was reviewed by members of the editorial board and by several independent reviewers. In light of the reviews (below this email), we would like to invite the resubmission of a significantly-revised version that takes into account the reviewers' comments. 

I must admit that your manuscript revision differed from what I had anticipated. Although you addressed some of the comments from reviewer 2 on a surface level, you chose to fully disregard the other comments made by this reviewer, as well as all the comments put forth by reviewer 1 (please see these comments below). 

Despite a valid foundation for rejection, we are extending an opportunity to you to provide responses to the previously unattended comments, along with more comprehensive explanations for the comments you initially tackled. Furthermore, kindly incorporate a more recent and widely accepted definition of mycetoma in your manuscript. Please take note that we expect your manuscript to undergo comprehensive adjustments that align entirely with your responses to the reviewers.

Reviewer 1

This is a single case

Was the M.mycetomatis identified by culture or by molecular methods ? 

Can you be more specific about the surgery ie was this visually complete excision. What doses of voriconazole were used and when was the patient changed to itraconazole

In Table 1 what does thrust mean ? Worsening ??

"The patient had to develop several previous pregnancies on this site of mycetoma, the outcome of which was favourable"

Does this mean that the patient had been pregnant several times before in the presence of the same mycetoma but had not developed any complications ? If so this isn’t what is said in the discussion where it is implied that the patients’ mycetoma was aggravated in all pregnancies

Chain Reactive Protein – C-reactive Protein

"Only a few cases have been reported in the literature concerning this association". I would be more careful here as you haven’t carried out a full literature review and there are more cases. So this is a real and significant association. I would change this to ….Cases have been reported in the literature concerning this association. You again say this is a rare occurrence in the discussion – I would rephrase this to say Although reported by several authors this is not a very common occurrence 

This author believes - there are 9 authors !

Voriconazole, first injected. Presumably this was by standard iv infusion rather than direct infiltration of the lesion

Table 2 

Is there a missing drug in the first column ? Also some of the data is repeated

Conclusion. You might point out that this drug cost problem is a very real one in many countries where the best management of Skin NTDs is compromised by the high cost of medications - in this case antifungals

Reviewer 2

The microbiologic identification of M. mycetomatis is not included

The authors do not explained the reason to stop antifungal therapy and changing to antibiotics

Table 1 is not necessary and should be substituted by a time lining 

Table 3 is not necessary

The authors should focus their discussion in the clinical and radiological outcome and especially the use of Antifungal drugs during pregnancy. 

In my opinion, advocacy related to therapy costs is non relevant.

The authors should change the focus of their manuscript.

They should discuss the antingungal drugs during pregnancy

The costs of antifungals and or the therapy in African countries is not scientifically interessante, but they could resume this issues in a single sentence

We cannot make any decision about publication until we have seen the revised manuscript and your response to the reviewers' comments. Your revised manuscript is also likely to be sent to reviewers for further evaluation.

Sincerely,

Marcio L Rodrigues

Section Editor

Marcio Rodrigues

Section Editor

I must admit that your manuscript revision differed from what I had anticipated. Although you addressed some of the comments from reviewer 2 on a surface level, you chose to fully disregard the other comments made by this reviewer, as well as all the comments put forth by reviewer 1 (please see these comments below). 

Despite a valid foundation for rejection, we are extending an opportunity to you to provide responses to the previously unattended comments, along with more comprehensive explanations for the comments you initially tackled. Furthermore, kindly incorporate a more recent and widely accepted definition of mycetoma in your manuscript. Please take note that we expect your manuscript to undergo comprehensive adjustments that align entirely with your responses to the reviewers.

Reviewer 1

This is a single case

Was the M.mycetomatis identified by culture or by molecular methods ? 

Can you be more specific about the surgery ie was this visually complete excision. What doses of voriconazole were used and when was the patient changed to itraconazole

In Table 1 what does thrust mean ? Worsening ??

"The patient had to develop several previous pregnancies on this site of mycetoma, the outcome of which was favourable"

Does this mean that the patient had been pregnant several times before in the presence of the same mycetoma but had not developed any complications ? If so this isn’t what is said in the discussion where it is implied that the patients’ mycetoma was aggravated in all pregnancies

Chain Reactive Protein – C-reactive Protein

"Only a few cases have been reported in the literature concerning this association". I would be more careful here as you haven’t carried out a full literature review and there are more cases. So this is a real and significant association. I would change this to ….Cases have been reported in the literature concerning this association. You again say this is a rare occurrence in the discussion – I would rephrase this to say Although reported by several authors this is not a very common occurrence 

This author believes - there are 9 authors !

Voriconazole, first injected. Presumably this was by standard iv infusion rather than direct infiltration of the lesion

Table 2 

Is there a missing drug in the first column ? Also some of the data is repeated

Conclusion. You might point out that this drug cost problem is a very real one in many countries where the best management of Skin NTDs is compromised by the high cost of medications - in this case antifungals

Reviewer 2

The microbiologic identification of M. mycetomatis is not included

The authors do not explained the reason to stop antifungal therapy and changing to antibiotics

Table 1 is not necessary and should be substituted by a time lining 

Table 3 is not necessary

The authors should focus their discussion in the clinical and radiological outcome and especially the use of Antifungal drugs during pregnancy. 

In my opinion, advocacy related to therapy costs is non relevant.

The authors should change the focus of their manuscript.

They should discuss the antingungal drugs during pregnancy

The costs of antifungals and or the therapy in African countries is not scientifically interessante, but they could resume this issues in a single sentence
---

## [Decision Letter · Decision Letter 2]

24 Sep 2023

Dear Dr. NDOUR,

Thank you very much for submitting your manuscript "Fungal mycetoma and pregnancy: an association with costly and difficult management, about a case" for consideration at PLOS Neglected Tropical Diseases. As with all papers reviewed by the journal, your manuscript was reviewed by members of the editorial board and by several independent reviewers. The reviewers appreciated the attention to an important topic. Based on the reviews, we are likely to accept this manuscript for publication, providing that you modify the manuscript according to the review recommendations. 

Thank you for your revision. Overall, I agree with reviewer 1; there have been improvements, but some English language editing is still needed. In Table 1 under "Thrust," we recommend changing it to "Worsening" in the table for clarity. It would also be beneficial to replace "exeresis" with "surgery" for better comprehension. Furthermore, please explicitly mention in the text that voriconazole was administered intravenously as the authors confirmed. This will enhance the clarity of your work.

Sincerely,

Marcio L Rodrigues

Section Editor

Marcio Rodrigues

Section Editor

Thank you for your revision. Overall, I agree with reviewer 1; there have been improvements, but some English language editing is still needed. In Table 1 under "Thrust," we recommend changing it to "Worsening" in the table for clarity. It would also be beneficial to replace "exeresis" with "surgery" for better comprehension. Furthermore, please explicitly mention in the text that voriconazole was administered intravenously as the authors confirmed. This will enhance the clarity of your work.

Reviewer's Responses to Questions

**Key Review Criteria Required for Acceptance?**

**Methods**

-Are the objectives of the study clearly articulated with a clear testable hypothesis stated?

-Is the study design appropriate to address the stated objectives?

-Is the population clearly described and appropriate for the hypothesis being tested?

-Is the sample size sufficient to ensure adequate power to address the hypothesis being tested?

-Were correct statistical analysis used to support conclusions?

-Are there concerns about ethical or regulatory requirements being met?

Reviewer #1: In general this is improved. It does need some editing to the English. 

I note the authors provided answers to my queries such as in Table 1 Thrust . This does mean Worsening - so it is important to change Thrust to Worsening in the table. I would also change exeresis to surgery

Likewise the authors confirm that the voriconazole was given intravenously - this should be stated in the text

Reviewer #2: (No Response)

**Results**

-Does the analysis presented match the analysis plan?

-Are the results clearly and completely presented?

-Are the figures (Tables, Images) of sufficient quality for clarity?

Reviewer #1: See above

Reviewer #2: (No Response)

**Conclusions**

-Are the conclusions supported by the data presented?

-Are the limitations of analysis clearly described?

-Do the authors discuss how these data can be helpful to advance our understanding of the topic under study?

-Is public health relevance addressed?

Reviewer #1: Yes

Reviewer #2: (No Response)

**Editorial and Data Presentation Modifications?**

Reviewer #1: See above

Reviewer #2: (No Response)

**Summary and General Comments**

Reviewer #1: See above

Reviewer #2: (No Response)

PLOS authors have the option to publish the peer review history of their article (what does this mean?). If published, this will include your full peer review and any attached files.

Reviewer #1: No

Reviewer #2: No

Figure Files:

Data Requirements:

Reproducibility:

References

---

## [Editor Report · Decision Letter 3]

9 Oct 2023

Dear Dr. NDOUR,

Thank you very much for submitting your manuscript "Fungal mycetoma and pregnancy: an association with costly and difficult management, about a case" for consideration at PLOS Neglected Tropical Diseases. As with all papers reviewed by the journal, your manuscript was reviewed by members of the editorial board and by several independent reviewers. The reviewers appreciated the attention to an important topic. Based on the reviews, we are likely to accept this manuscript for publication, providing that you modify the manuscript according to the review recommendations. 

Thank you for your manuscript's revision. Reviewer 1 still has minor concerns - please address them and return the revised manuscript to us.

Sincerely,

Marcio L Rodrigues

Section Editor

Marcio Rodrigues

Section Editor

Thank you for your manuscript's revision. Reviewer 1 still has minor concerns - please address them and return the revised manuscript to us.

Figure Files:

Data Requirements:

Reproducibility:

References

---

## [Editor Report · Decision Letter 4]

24 Oct 2023

Dear Dr. NDOUR,

We are pleased to inform you that your manuscript 'Fungal mycetoma and pregnancy: an association with costly and difficult management, about a case' has been provisionally accepted for publication in PLOS Neglected Tropical Diseases.

Best regards,

Marcio L Rodrigues

Section Editor

Marcio Rodrigues

Section Editor

---

## [Editor Report · Acceptance letter]

5 Nov 2023

Dear Dr. NDOUR,

We are delighted to inform you that your manuscript, "Fungal mycetoma and pregnancy: an association with costly and difficult management, about a case," has been formally accepted for publication in PLOS Neglected Tropical Diseases.

Best regards,

Shaden Kamhawi

co-Editor-in-Chief

Paul Brindley

co-Editor-in-Chief
